# Smoking and High-Sensitivity Troponin I Levels in Young and Healthy Adults from the General Population

Kirsten Grossmann [1,2], Stefanie Aeschbacher [3], Niklas Wohlwend [1], Steffen Blum [3,4], Ornella C. Weideli [1], Julia Telser [2,5], Martin Risch [1,6], Lorenz Risch [1,2,7] and David Conen [4,*]

1   Dr Risch Medical Laboratory, 9490 Vaduz, Liechtenstein
2   Faculty of Medical Sciences, Private University Liechtenstein, 9495 Triesen, Liechtenstein
3   Cardiovascular Research Institute Basel and Division of Cardiology, University Hospital Basel, University of Basel, 4031 Basel, Switzerland
4   Population Health Research Institute, McMaster University, Hamilton, ON L8L 2X2, Canada
5   Dr Risch Medical Laboratory, 9470 Buchs, Switzerland
6   Division of Laboratory Medicine, Cantonal Hospital Graubünden, 7007 Chur, Switzerland
7   Department of Laboratory Medicine, Institute of Clinical Chemistry, Inselspital Bern University Hospital, University of Berne, 3012 Berne, Switzerland
*   Correspondence: david.conen@phri.ca or conend@mcmaster.ca

**Abstract:** Lower troponin concentrations measured in smokers in a healthy population raise the question of whether a lower troponin threshold should be considered for tobacco users. We aim to evaluate differences in troponin levels according to the smoking status in healthy young adults. Participants aged 25–41 years were enrolled in a population-based observational study. The smoking status was self-assessed, and participants were classified as never-, past-, and current smokers. Pack-years of smoking were calculated. High-sensitivity cardiac troponin I (hs-cTnI) concentrations were measured from thawed blood samples, and associations were assessed using multivariable linear regression analyses. We included 2155 subjects in this analysis. The mean (SD) age was $35.4 \pm 5.22$ years; 53% were women. The median hs-cTnI levels across smoking status categories were 0.70 (interquartile range 0.43–1.23) ng/L in never smokers (n = 1174), 0.69 (interquartile range 0.43–1.28) ng/L in past smokers (n = 503), and 0.67 (interquartile range 0.41–1.04) ng/L in current smokers (n = 478), $p = 0.04$. The troponin levels remained significantly lower in current smokers after adjustment for potential confounders (β-coefficient [95%CI] of $-0.08$ [$-0.25$; $-0.08$], $p < 0.001$). Our results confirm that current smokers have lower hs-cTnI levels than past or never smokers, with a significant dose–response relationship among current smokers. The absolute differences in hs-cTnI levels were small.

**Keywords:** biomarkers; cardiovascular system; population; smoking; Troponin I

## 1. Introduction

High-sensitivity cardiac troponin (hs-cTn) assays can be used to diagnose acute myocardial infarction and also to detect individuals at high risk of having future cardiovascular events [1–3]. Previous data in patients with stable coronary arterial disease (CAD) demonstrated that cigarette smoking was associated with lower concentrations of circulating high-sensitivity cardiac troponin T (hs-cTnT); however, no such relationship was found for high-sensitivity cardiac troponin I (hs-cTnI) [4]. Significant associations between hs-cTnT and smoking were also found in a large community-based cohort free of CAD [5]. Evidence indicates lower levels of hs-cTnI concentrations in current smokers (19 to 94 years old) without known cardiovascular disease or diabetes mellitus [6,7]. The results from the Multi Ethnic Study of Atherosclerosis demonstrated that the intensity of smoking may be more related to cardiovascular disease (CVD) events than the duration [8]. There are multiple reasons why smoking could be associated with lower troponin levels, including survival

bias, residual confounding, or interference in the assay used to measure cTn levels. Different levels of cardiac troponins among smokers versus non-smokers may have implications for diagnostic and prognostic purposes. The association between hs-cTnI and smoking is controversial, and a better understanding of this relationship is needed. For instance, assessing the association in young and healthy individuals may minimize some concerns related to the above-mentioned bias. Furthermore, the impact of smoking intensity on hs-cTnI levels has not been investigated previously and may provide clues about the causality of the association. The aim of this study was to investigate the relationships of the smoking status, number of cigarettes smoked, and smoking intensity with hs-cTnI levels in young (25–41 years old) and healthy adults from the general population. Accordingly, we hypothesized that smoking would be associated with lower concentrations of hs-cTnI.

## 2. Materials and Methods

### 2.1. Study Participants

The current analysis was performed within the GAPP (genetic and phenotypic determinants of blood pressure and other cardiovascular risk factors) study. GAPP is an ongoing prospective population-based cohort study among initially healthy adults in the Principality of Liechtenstein. Participant selection methods have been previously described [9,10]. In brief, 2170 participants (25–41 years old) living in the Principality of Liechtenstein were enrolled between 2010 and 2013. The main exclusion criteria were established cardiovascular disease, body mass index >35 kg/m², diagnosed sleep apnea syndrome, intake of antidiabetic drugs, and any other severe illnesses. The local ethics committee (KEK, Zürich, Switzerland) approved the study protocol, and written informed consent was obtained from each participant. Of the 2170 participants enrolled in GAPP, some were excluded due to missing data on their smoking status (n = 3) or missing hs-cTnI values (n = 11); an outlier (n = 1) with an apparent hs-cTnI value of >1000 ng/L was also excluded, resulting in 2155 participants eligible for this analysis (Figure 1).

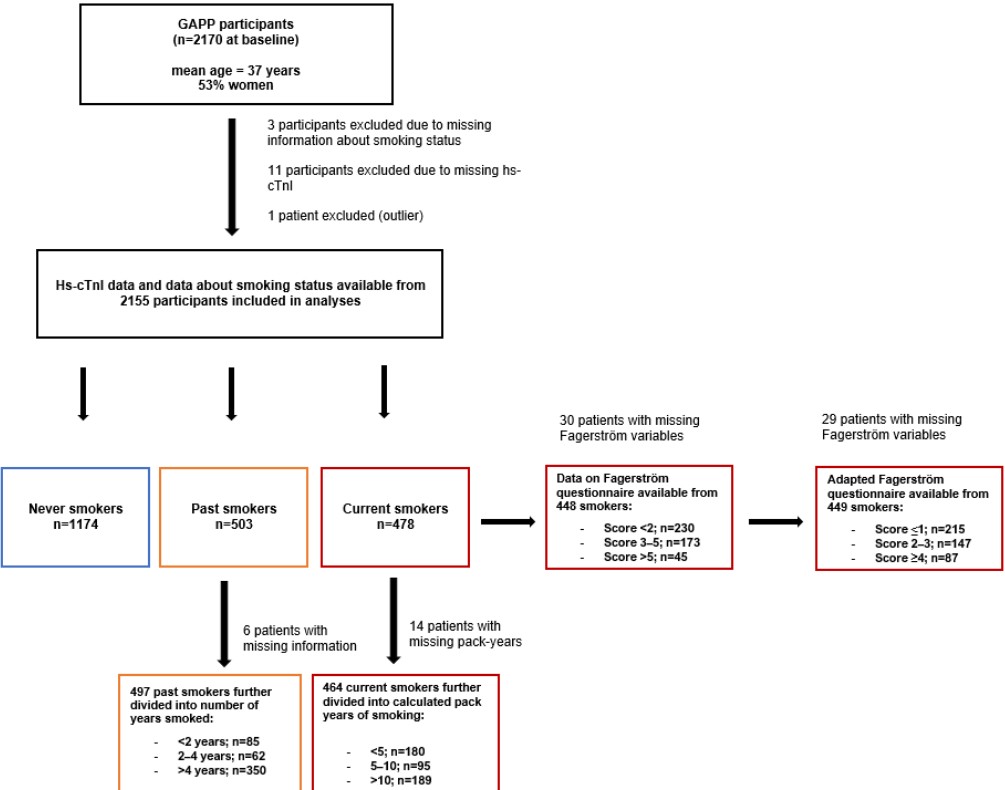

**Figure 1.** Flowchart of troponin and smoking data.

### 2.2. Assessment of Smoking

Smoking information was self-assessed between 2010 and 2013 using a detailed questionnaire. The smoking status was classified as current (having smoked > 100 lifetime cigarettes and currently smoke cigarettes), past (having smoked > 100 lifetime cigarettes but had quit smoking at the time of the questionnaire), or never (having smoked < 100 lifetime cigarettes) smoking. Past smokers reported the year of starting and stopping smoking, whereas current smokers reported the year of starting smoking. Current smokers also reported the average number of cigarettes they smoked per day, allowing the calculation of pack-years (i.e., product of the number of years of smoking with the average number of cigarettes smoked per day). In addition, current smokers completed the validated Fagerström questionnaire [11,12], used to quantify smoking intensity/nicotine dependence. Based on the Fagerström questions, a score between 0 and 10 was calculated and smokers were further assigned to three pre-specified groups (2, 3–5 and >5) where higher scores indicate higher smoking intensity [13,14].

### 2.3. Assessment of Troponin and Other Laboratory Parameters

Fasting venous blood samples were drawn from each study participant and immediately stored at $-80^{\circ}$C after centrifugation. Hs-cTnI was determined from thawed EDTA plasma samples using a high-sensitivity, single-molecule counting assay (Erenna Immunoassay System, Singulex, Alameda, CA, USA) with a 0.04 ng/L limit of detection, and intra-/inter-assay coefficients with a variation of 6%/6% at a cTnI concentration of 79.7 ng/L [15]. Eleven individuals with undetectable cTnI levels were assigned a cTnI value of 0.04 ng/L. Additional biomarkers measured directly from fresh blood samples were low-density lipoprotein cholesterol (LDL-C in mml/L), high-density lipoprotein cholesterol (HDL-C in mmol/L), and total cholesterol.

### 2.4. Assessment of Further Study Variables

Standardized questionnaires were used to assess information about personal, medical, and lifestyle factors. Education was documented by noting the highest achieved education level. Regular physical activity was defined as >150 min of vigorous activity per week and was evaluated using the validated International Physical Activity Questionnaire (IPAQ) [16]. Regular fruit and vegetable consumption were defined as consuming at least five servings per day. Alcohol consumption was dichotomized, with participants stratified into drinkers and non-drinkers. Height and body weight were directly measured in a standardized manner [9]. Body mass index (BMI) was calculated as weight in kilograms over height in meter squared. Ambulatory 24 h blood pressure measurements were obtained using a validated automatic device (BR-102 plus; Schiller AG, Baar, Switzerland), and the mean over 24 h was used for this analysis. Hypertension was defined as systolic blood pressure $\geq 140$ mmHg, diastolic blood pressure $\geq 90$ mmHg, or intake of blood-pressure-lowering drugs.

### 2.5. Statistical Analysis

All statistical analyses were performed using the statistical software IBM SPSS Statistics Version 26 [17]. Baseline characteristics were stratified by smoking status (never, current, past). Continuous data are presented as mean ($\pm$standard deviation) or median (interquartile range) and compared using analysis of variance (ANOVA) or Kruskal–Wallis tests as appropriate. Categorical variables are presented as counts (percentages) and compared with Chi-square tests (Table 1).

**Table 1.** Baseline characteristics according to smoking status.

| N = 2155 | Never Smokers N = 1174 | Past Smokers N = 503 | Current Smokers N = 478 | *p*-Value |
|---|---|---|---|---|
| Sex, Female (%) | 667 (58.1) | 260 (51.7) | 222 (46.4) | <0.001 |
| Age, Years | 35.6 ± 5.3 | 35.8 ± 4.9 | 34.5 ± 5.3 | <0.001 |
| Height, cm | 172 ± 9.19 | 172 ± 9.09 | 172 ± 9.11 | 0.31 |
| Weight, kg | 71 ± 14 | 75 ± 16 | 75 ± 15 | <0.001 |
| BMI, kg/m$^2$ | 24 ± 4 | 25 ± 4 | 25 ± 4 | <0.001 |
| Physical Activity [1] (%) | 619 (52.7) | 270 (53.7) | 244 (51.0) | 0.70 |
| Alcohol Consumption, g/24 h | 0.00 (0.00; 1.44) | 0.80 (0.00; 2.14) | 0.80 (0.00; 2.35) | <0.001 |
| Highest Education Level (%) | 504 (43.4) | 175 (35.10) | 117 (24.8) | <0.001 |
| Fruit/Vegetable Consumption [2] (%) | 271 (23.1) | 110 (21.9) | 82 (17.2) | 0.03 |
| Systolic BP, mm Hg/24 h | 122.27 ± 11.53 | 124.91 ± 11.00 | 124.41 ± 11.89 | <0.001 |
| Diastolic BP, mm Hg/24 h | 78.15 ± 7.94 | 79.35 ± 8.06 | 79.53 ± 8.30 | <0.001 |
| Hypertension (%) | 163 (13.9) | 80 (15.9) | 59 (12.3) | 0.27 |
| LDL-C, mmol/L | 2.84 (2.35; 3.44) | 2.85 (2.31; 3.44) | 2.98 (2.38; 3.63) | 0.06 |
| HDL-C, mmol/L | 1.53 (1.30; 1.84) | 1.50 (1.22; 1.80) | 1.37 (1.11; 1.66) | <0.001 |
| Total Cholesterol, mmol/L | 4.81 (4.26; 5.48) | 4.78 (4.29; 5.35) | 4.78 (4.27; 5.52) | 0.81 |
| Hs-CTnI, ng/L | 0.70 (0.43; 1.23) | 0.69 (0.43; 1.28) | 0.67 (0.41; 1.04) | 0.04 |

Data are presented as mean ± SD, median (interquartile range) or number (percentage). [1] Regular physical activity was defined as >150 min of vigorous activity per week. [2] Regular consumption of fruits and/or vegetables was defined as 5 servings per day.

To investigate the associations between smoking and hs-cTnI levels, we constructed multivariable adjusted linear regression models using log-transformed hs-cTnI levels as the dependent variable. Hs-cTnI levels were log-transformed due to their skewed distribution. In a first model (Table 2), the smoking status was used as the predictor of interest. In a second model (Table 3), a potential dose–response relationship was assessed using the number of pack-years (classified as <5, 5–10 or >10) as the predictor of interest. Based on the Fagerström test score [13,14], current smokers were allocated to three groups (2, 3–5, and >5 points) to further assess the relationship between smoking intensity and hs-cTnI levels (Table 4). To investigate the association between years since smoking cessation and hs-cTnI levels, years since smoking cessation were classified as <2 years, 2–4 years and >4 years. (Table 5). The never-smoking group was used as reference for analysis performed using the linear regression models. All the models were first adjusted for age and sex. In a second step, the models were additionally adjusted for a predefined set of covariates including BMI, hypertension, alcohol consumption, educational status, fruit/vegetable consumption, physical activity, LDL-C, HDL-C, and total cholesterol.

**Table 2.** Association between smoking status and hs-cTnI.

| N = 2155 | Never Smokers N = 1174 | Past Smokers N = 503 | Current Smokers N = 478 | *p*-Value for Trend |
|---|---|---|---|---|
| Univariate | Ref | 0.01 (−0.07; 0.12), $p = 0.54$ | −0.05 (−0.21; −0.02), $p = 0.02$ | $p = 0.03$ |
| Sex and age adjusted | Ref | −0.01 (−0.10; 0.07), $p = 0.69$ | −0.09 (−0.28; −0.11), $p < 0.001$ | $p < 0.001$ |
| Multivariable adjusted [1] | Ref | −0.02 (−0.12; 0.04), $p = 0.34$ | −0.08 (−0.25; −0.08), $p < 0.001$ | $p < 0.001$ |

Data are presented as β-coefficient (95% confidence interval). High sensitivity Troponin I was log-transformed. [1] Adjusted for sex, age, body mass index, hypertension, alcohol consumption, educational status, fruit/vegetable consumption, physical activity, low density lipoprotein cholesterol, high density lipoprotein cholesterol, total cholesterol.

**Table 3.** Pack-years of smoking in relation to hs-cTnI.

|  | Never Smokers N = 1174 | Past Smokers N = 503 | <5 Pack-Years N = 180 | 5–10 Pack-Years N = 95 | >10 Pack-Years N = 189 | *p*-Value for Trend |
|---|---|---|---|---|---|---|
| Hs-cTnI [1] | 0.71 [0.30–1.10] | 0.69 [0.26–1.12] | 0.63 [0.31–0.94] | 0.66 [0.31–1.02] | 0.69 [0.42–0.95] | *p* = 0.26 |
| Univariate | Ref | 0.01 (−0.06; 0.12), *p* = 0.53 | −0.03 (−0.24; 0.04), *p* = 0.15 | −0.03 (−0.30; 0.08), *p* = 0.25 | −0.04 (−0.26; 0.02), *p* = 0.09 | *p* = 0.14 |
| Sex and age adjusted | Ref | −0.01 (−0.10; 0.07), *p* = 0.74 | −0.03 (−0.24; 0.01), *p* = 0.07 | −0.03 (−0.29; 0.04), *p* = 0.15 | −0.09 (−0.40; −0.16), *p* < 0.001 | *p* < 0.001 |
| Multivariable adjusted [2] | Ref | −0.02 (−0.12; 0.04), *p* = 0.37 | −0.03 (−0.23; 0.02), *p* = 0.09 | −0.02 (−0.27; 0.06), *p* = 0.21 | −0.08 (−0.36; −0.11), *p* < 0.001 | *p* < 0.001 |

Data are presented as β-coefficient (95% confidence interval). High sensitivity Troponin I was log-transformed. [1] Data are median [25–75%]. [2] Adjusted for sex, age, body mass index, hypertension, alcohol consumption, educational status, fruit/vegetable consumption, physical activity, low density lipoprotein cholesterol, high density lipoprotein cholesterol, total cholesterol.

**Table 4.** Results of the relationship between Fagerström scores and hs-cTnI.

| N = 2155 | Never Smokers N = 1174 | Past Smokers n = 503 | Current Smokers N = 448 | | | *p*-Value for Trend |
|---|---|---|---|---|---|---|
|  |  |  | Fagerström-Score <2 N = 230 | Fagerström-Score 3–5 N = 173 | Fagerström-Score >5 N = 45 |  |
| Hs-cTnI [1] | 0.71 (0.80) [0.30–1.10] | 0.69 (0.86) [0.26–1.12] | 0.68 (0.70) [0.34–1.02] | 0.66 (0.59) [0.35–0.94] | 0.66 (0.50) [0.41–0.91] | *p* = 0.14 |
| Univariate | Ref. | 0.01 (−0.07; 1.12), *p* = 0.59 | −0.02 (−0.18; 0.06), *p* = 0.35 | −0.06 (−0.33; −0.04), *p* = 0.013 | −0.04 (−0.51; 0.02), *p* = 0.07 | *p* = 0.03 |
| Sex and age adjusted | Ref. | 0.01 (−0.10; 0.07), *p* = 0.68 | −0.03 (−0.20; 0.02), *p* = 0.10 | −0.09 (−0.41; −0.16), *p* < 0.001 | −0.07 (−0.66; −0.19), *p* < 0.001 | *p* < 0.001 |
| Multivariable adjusted [2] | Ref. | −0.02 (−0.12; 0.04), *p* = 0.33 | −0.03 (−0.19; 0.03), *p* = 0.17 | −0.08 (−0.38; −0.12), *p* < 0.001 | −0.06 (−0.61; −0.13), *p* < 0.001 | *p* < 0.001 |

Data are presented as β-coefficient (95% confidence interval). High sensitivity Troponin I was log-transformed. [1] Data are median [25–75%]. [2] Adjusted for sex, age, body mass index, hypertension, alcohol consumption, educational status, fruit/vegetable consumption, physical activity, low density lipoprotein cholesterol, high density lipoprotein cholesterol, total cholesterol.

**Table 5.** Results of the relationship between smoking cessation and hs-cTnI.

| N = 2149 | Never Smokers N = 1174 | Current Smokers N = 478 | Past Smokers N = 497 | | | *p*-Value for Trend |
|---|---|---|---|---|---|---|
|  |  |  | <2 years N = 85 | 2–4 Years N = 62 | >4 Years N = 350 |  |
| Hs-cTnI [1] Median (IQR) [25–75%] | 0.71 (0.80) [0.30–1.10] | 0.67 (0.62) [0.36–0.98] | 0.75 (1.11) [0.21–1.28] | 0.74 (0.63) [0.42–1.05] | 0.66 (0.87) [0.21–1.08] | *p* = 0.17 |
| Univariate | Ref. | −0.05 (−0.21; −0.02), *p* = 0.02 | 0.02 (−0.11; 0.28), *p* = 0.41 | 0.01 (−0.16; 0.30), *p* = 0.61 | 0.004 (−0.10; 0.12), *p* = 0.86 | *p* = 0.094 |
| Sex and age adjusted | Ref. | −0.09 (−0.28; −0.11), *p* = 0.001 | −0.01 (−0.21; 0.14), *p* = 0.72 | 0.01 (−0.14; 0.27), *p* = 0.52 | −0.01 (−0.12; 0.07), *p* = 0.55 | *p* = 0.001 |
| Multivariable adjusted [2] | Ref. | −0.08 (−0.25; −0.08), *p* = 0.001 | −0.01 (−0.21; 0.14), *p* = 0.67 | 0.01 (−0.15; 0.26), *p* = 0.58 | −0.02 (−0.15; 0.04), *p* = 0.23 | *p* = 0.001 |

Data are presented as β-coefficient (95% confidence interval). High sensitivity Troponin I was log-transformed. [1] Data are median [25–75%]. [2] Adjusted for sex, age, body mass index, hypertension, alcohol consumption, educational status, fruit/vegetable consumption, physical activity, low density lipoprotein cholesterol, high density lipoprotein cholesterol, total cholesterol.

All statistical analyses were performed using IBM SPSS Statistics Version 26. A two-tailed *p*-value < 0.05 was used to indicate statistical significance.

There was no patient or public involvement.

## 3. Results

Baseline characteristics stratified by smoking status are presented in Table 1.

Multiple significant between-group differences were observed. For example, the mean (SD) age in current smokers was 34.5 ± 5.3 years; in past smokers, 35.8 ± 4.9 years; and in

never smokers, 35.6 ± 5.3 years. The prevalence in women was 58.1%, 51.7%, and 46.4% among never-, past-, and current smokers, respectively ($p < 0.001$). The median hs-cTnI levels were lower in the current smokers (0.67 ng/L) compared to that in never smokers (0.70 ng/L), and past smokers (0.69 ng/L), $p = 0.04$.

Results from the linear regression analyses on the associations between smoking status and log-transformed hs-cTnI levels are shown in Table 2.

Compared to never smokers, current smokers had significantly lower hs-cTnI levels (β-coefficient [95%CI] of −0.05 [−0.21; −0.02], $p = 0.02$); however, this difference was not statistically significant when compared with that for past smokers (β-coefficient [95%CI] of 0.14 [−0.07; −0.13], $p = 0.54$). After comprehensive adjustment for potential confounders, current smokers still had lower hs-cTnI levels compared to never smokers (β-coefficient [95%CI] −0.08 [−0.25; −0.08], $p = <0.001$). Again, no significant difference was observed between never smokers and past smokers (β-coefficient [95%CI] of −0.02 [−0.12; 0.05], $p = 0.37$).

Analyses of the relationship between smoking intensity and log-transformed hs-cTnI levels are shown in Table 3.

After adjustment for sex and age, smokers with >10 pack years had significantly lower hs-cTnI levels than never smokers (β-coefficient [95%CI] −0.09 [−0.40; −0.16], $p < 0.001$). Results remained very similar after additional multivariable adjustment (β-coefficient [95%CI] −0.08 [−0.36; −0.11], $p < 0.001$). The $p$ value for trend across different categories of smoking intensity was highly significant ($p < 0.001$).

Analyses of the relationship between Fagerström score and log-transformed hs-cTnI levels are shown in Table 4.

Compared to never smokers, all the Fagerström groups presented lower hs-cTnI levels, with a significant $p$-value for trend across different categories.

Analysis between the years of smoking cessation and hs-cTnI levels in the group of past smokers is shown in Table 5.

Past smokers were further divided into three groups according to the number of years not smoked. In all adjusted models no significant difference in hs-cTnI levels of past smokers was found compared to never smokers.

## 4. Discussion

In our large population-based study of young and healthy adults, we found significantly lower hs-cTnI concentrations in current smokers compared to that in never smokers. A large difference was observed in participants with a high life-time consumption of cigarettes and a high smoking intensity. Past smokers had similar hs-cTnI levels to never smokers. The absolute between-group differences in hs-cTnI concentrations were small.

Our findings are in line with and expand previously published data showing lower levels of hs-cTnI in smokers compared to those in never smokers in the general population [7]. Previous research on hs-cTnI levels in smokers showed results with similar small deviations between smoking groups aged 19 to 94 exhibiting significantly lower levels of cTnI than never smokers and former smokers [6]. However, data in stable CAD from smokers compared to non-smokers presented significantly lower hs-cTnT but not hs-cTnI concentrations [4].

Compared to the existing literature investigating hs-cTnI across a wide range of age groups, our results demonstrate lower hs-cTnI levels in adults aged 25 to 41 who currently smoke. Analyzing hs-cTnI in this narrow young age group minimizes the risk of unmeasured confounding or survival bias and also lowers the risk of undetected CAD. Our study also adds to the existing literature by showing the intensity dose–response relationship by the used Fagerström score.

Like previous studies [4,6,7], we evaluated hs-cTnI concentrations in different smoking groups (current, past, and never) and showed lower hs-cTnI levels in current smokers compared to those in never smokers. We also calculated pack-years to measure the lifetime exposure to smoking and found a strong trend across pre-specified groups of pack-years

smoked. Smoking intensity as measured by the validated Fagerström score [13] was also inversely associated with troponin levels. Our results, therefore, provide evidence of a dose–response relationship, suggesting that the observed associations may be causal.

Our study further confirms the results of prior studies showing smoking as one of the main cardiovascular risk factors associated with lower hs-cTnI levels [6,7]. Our results underline these findings for a group of young (25–41 years old) and healthy adults from the general population The level of cardiac troponin hs-cTnI is low in the studied population of smokers, and there are no significant differences in troponin values between the compared groups. These findings are somewhat paradoxical since smoking is a strong cardiovascular risk factor and higher troponin levels are also associated with a worse cardiovascular risk profile and worse cardiovascular outcomes [1,18]. However, given the known adverse health effects of smoking, it is very unlikely that this association reflects a protective effect. More likely, negative feedback mechanisms produced by smoking could be a mechanistic for the perception of low hs-cTnI levels in circulation that induce an increase in its production at the level of myocardium, causing damage to the myocardial cell. Additionally, other mechanisms could be involved that will need more clarification in future investigations. Whether the small differences observed between groups should lead to different reference value ranges of hs-cTn concentrations is currently unclear and should be evaluated in future studies [19–21]. Given the small differences and the fact that most routine laboratories cannot detect these troponin levels, we expect a small clinical impact.

The strengths of our study are the large sample size of well-characterized young and healthy adults with a relatively short life-time exposure to environmental confounders within a population-based study design. The potential limitations of our dataset include its cross-sectional design, the fact that most of the study population was white, and the fact that all smoking information was self-reported. These hamper the widespread generalizability of our results. Randomized controlled clinical trials would be necessary to assess whether smoking intensity lowers hs-cTnI levels.

In conclusion, hs-cTnI levels were lower among current smokers compared to those in never- and past smokers in a large sample of young and healthy adults. We found a dose–response relationship between pack-years of smoking and smoking intensity. The absolute differences in hs-cTnI levels between groups were small.

**Author Contributions:** All the authors critically reviewed and approved the final version of this manuscript and had final responsibility for the decision to submit for publication. Conceptualization: M.R., L.R. and D.C.; data curation: K.G. and S.A. formal analysis: K.G. and S.A.; funding acquisition: M.R., L.R. and D.C.; investigation: K.G. and S.A.; methodology: M.R., L.R. and D.C.; project administration: M.R., K.G., S.A., D.C. and L.R.; resources: M.R., K.G., S.A., D.C. and L.R.; supervision: M.R., S.A., D.C. and L.R.; validation: M.R., K.G., S.A., D.C., S.B. and L.R.; visualization: K.G.; writing—original draft: K.G.; writing—review and editing: M.R., K.G., S.A., O.C.W., J.T., N.W., S.B., L.R. and D.C. All authors have read and agreed to the published version of the manuscript.

**Funding:** The GAPP study was supported by the Liechtenstein Government, the Swiss National Science Foundation, the Swiss Heart Foundation, the Swiss Society of Hypertension, the University of Basel, the University Hospital Basel, the Hanela Foundation, Schiller AG, and Novartis.

**Institutional Review Board Statement:** The local ethics committee (KEK, Zürich, Switzerland; EK 66/09) approved the study protocol, and written informed consent was obtained from each participant.

**Informed Consent Statement:** Informed consent was obtained from all subjects involved in the study.

**Data Availability Statement:** Anonymized data that underlie the results reported in this article are available upon reasonable request to the corresponding author.

**Acknowledgments:** The authors thank the participants and staff of the GAPP study for their important contributions.

**Conflicts of Interest:** David Conen has received consulting fees from Roche Diagnostics, outside of the current work. Lorenz Risch, and Martin Risch are key shareholders of the Dr Risch Medical Laboratory. The other authors have no financial or personal conflicts of interest to declare.

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
