# Peer review of "Smoking and High-Sensitivity Troponin I Levels in Young and Healthy Adults from the General Population"

_applsci, doi:10.3390/app12199777_

Round 1

Reviewer 1 Report

The article titled “Smoking and high-sensitivity troponin I levels in young and healthy adults from the general population” by Grossmann et al. investigates the associations between status/intensity of smoking and high-sensitivity cardiac troponin I (hs-cTnI) levels in young (25-41 year old) and healthy adults. The authors have determined hs-cTnI levels in blood samples and the smoking status of participants were self-reported. Overall, the study provides a detailed comparison of hs-cTnI levels in a relatively large group of healthy adults based on their smoking status.

Similar hs-cTnI levels are observed for past and never smokers. The paper would be stronger if how long it has been since the participants quit smoking at the time of the questionnaire is also reported.

Reviewer 2 Report

I carefully read the article resulting from your study and found that it is very well done from the point of view of the number of patients included, the material studied and the methods used, as well as from the point of view of the statistical analysis. Thus, the results obtained are reliable and are the product of correctly applied statistical tests on the values ​​of the studied parameters.

In the light of the obtained results, it would be desirable for the discussions on these results to focus on the fact that the level of cardiac troponin hs-cTnI is low in the studied population of smokers, as well as on the fact that there are no significant differences in troponin values ​​between the compared groups.

This very result, which intrigued you, involves discussions on several levels.

If troponin is low in this category of individuals included in the study, is smoking a protective factor? Not likely considering what is known today regarding the cardiovascular risk induced by smoking.

If cardiac troponin is low in smokers, can we talk about a negative feed-back mechanism produced by smoking? In this case, the perception of a low level of cardiac troponin in circulation induces an increase in its production at the level of the myocardium, and would this increase itself cause damage to the myocardial cell? I think these aspects should be discussed more widely.
